# Gestational Age-Dependent Regulation of Transthyretin in Mice during Pregnancy

**DOI:** 10.3390/biology12081048

**Published:** 2023-07-26

**Authors:** Shibin Cheng, Zheping Huang, Akitoshi Nakashima, Surendra Sharma

**Affiliations:** 1Department of Pediatrics, Women & Infants Hospital, Rhode Island and Brown University, Providence, RI 02905, USA; hzpsippr3@gmail.com; 2Department of Obstetrics and Gynecology, Faculty of Medicine, University of Toyama, Toyama 930-8555, Japan; akinaka@med.u-toyama.ac.jp

**Keywords:** transthyretin, placenta, pregnancy, IL-10, IL-10 knockout mice, protein aggregation, liver

## Abstract

**Simple Summary:**

Preeclampsia is a severe pregnancy complication with a high incidence of maternal and fetal morbidity and mortality. Our work has demonstrated that preeclampsia shares the etiology of protein misfolding and aggregation with neurodegenerative diseases such as Alzheimer’s. We identified that transthyretin, a transporter of thyroxine and retinol, was a key component in the protein complex in the preeclampsia placenta. However, how TTR is regulated during pregnancy is not clear. In this study, we show that transthyretin is temporally downregulated in the liver and the placenta in pregnant mice and that inflammation may further orchestrate its further downregulation. Importantly, the levels of transthyretin are normalized postpartum. This research provides important insights into understanding the regulation and function of TTR in pregnancy.

**Abstract:**

Our prior studies have shown that protein misfolding and aggregation in the placenta are linked to the development of preeclampsia, a severe pregnancy complication. We identified transthyretin (TTR) as a key component of the aggregated protein complex. However, the regulation of native TTR in normal pregnancy remains unclear. In this study, we found that pregnant mice exhibited a remarkable and progressive decline in serum TTR levels through gestational day (gd) 12–14, followed by an increase in late pregnancy and postpartum. Meanwhile, serum albumin levels showed a modest but statistically significant increase throughout gestation. TTR protein and mRNA levels in the liver, a primary source of circulating TTR, mirrored the changes observed in serum TTR levels during gestation. Intriguingly, a similar pattern of TTR alteration was also observed in the serum of pregnant women and pregnant interleukin-10-knockout (IL-10^−/−^) mice with high inflammation background. In non-pregnant IL-10^−/−^ mice, serum TTR levels were significantly lower than those in age-matched wild-type mice. Administration of IL-10 to non-pregnant IL-10^−/−^ mice restored their serum TTR levels. Notably, dysregulation of TTR resulted in fewer implantation units, lower fetal weight, and smaller litter sizes in human TTR-overexpressing transgenic mice. Thus, TTR may play a pivotal role as a crucial regulator in normal pregnancy, and inflammation during pregnancy may contribute to the downregulation of serum TTR presence.

## 1. Introduction

Transthyretin (TTR) is a homotetrameric plasma protein of 55 kDa, originally designated as pre-albumin in 1942. It is primarily synthesized in the liver and the choroid plexus, with lower levels in the visceral yolk sac, placenta, retina, pancreas, intestine, stomach, heart, and spleen [1,2,3,4]. TTR is also expressed in the trophoblasts of the human placenta as early as 6 weeks of pregnancy and throughout gestation [2,3]. TTR is known for its role as a minor transporter of thyroxine and the major carrier of retinol-binding protein charged with retinol in the plasma, but accumulating evidence has revealed additional roles in metabolism, growth, fertility, depression-like behavior, angiogenesis, memory capacities during aging, and nerve regeneration after injury [5,6,7,8]. TTR also functions as a chaperone to protect neurons against the toxicity of amyloid-beta oligomers in transgenic murine models of Alzheimer’s disease [9,10]. Moreover, TTR misfolding and aggregation have been shown to lead to the development of senile systemic amyloidosis, familial amyloidotic polyneuropathy, and cardiomyopathy [11,12,13].

Our prior studies have demonstrated that TTR misfolding and aggregation contribute to the pathogenesis of preeclampsia (PE), a pregnancy-specific, multifactorial syndrome characterized by hypertension after 20 weeks of gestation [14,15,16,17,18,19,20,21]. PE is a leading cause of maternal and fetal morbidity and mortality, affecting 3–8% of all pregnancies worldwide [14,15,16,17,18,19,20,21]. Despite extensive investigations, the etiology of PE remains poorly understood. We showed that injection of sera from PE patients (PES) into pregnant mice induced PE-like features, which could be blocked if aggregated TTR was depleted from PES using a specific antibody [16,22]. This is further corroborated by the observations that the administration of serum immunoprecipitates containing TTR aggregates caused PE-like features in pregnant mice with a higher severity index in IL-10^−/−^ mice [16]. In addition, aggregated TTR is a component of the cargo of nano-vesicles released from PE placental explants, which are likely to act as damage-associated molecular pattern molecules in maternal circulation, leading to sterile inflammation associated with the PE pathology [20,23,24,25]. Our recent studies have revealed a robust deposition of TTR aggregates in trophoblasts in the PE placenta. This is thought to be caused by excessive endoplasmic reticulum (ER) stress, exhausted unfolded protein response, and impaired autophagy-lysosomal machinery [17,18]. Furthermore, transgenic mice over-expressing human TTR experienced TTR aggregate accumulation in the placenta, PE-like features, and increased production of sFlt-1 and soluble endoglin [17]. Taken together, our prior findings indicate TTR aggregation as a causative contributor to the PE etiology.

The role of TTR in maintaining normal pregnancy and how it is regulated during pregnancy remains to be elucidated. In this study, we show that when compared to non-pregnant mice, TTR presence in the sera and placenta of mice remarkably declines, with the lowest levels observed at gestational day (gd) 12–14, a pattern that parallels TTR mRNA levels in the liver. Interestingly, IL-10^−/−^ mice with high intrinsic inflammation background have lower levels of serum TTR compared to wild-type mice, and the administration of IL-10 protein restores physiological serum TTR levels in IL-10^−/−^ mice. Notably, overexpression of human TTR disturbs normal pregnancy. Thus, our results provide important insights into the role of TTR in normal pregnancy.

## 2. Materials and Methods

### 2.1. Animals and Treatment

C57BL/6 wild-type (n = 80) and IL-10^−/−^ mice (n = 50) were purchased from The Jackson Laboratory (Bar Harbor, ME, USA). Transgenic mice over-expressing human TTR (huTTR mice, n = 22) were provided as a gift from Dr. Joel N. Buxbaum. The transgene construct contains 90–100 copies of the normal human TTR gene and all the known regulatory elements required for tissue-specific expression. All animal protocols were approved by the Lifespan Institutional Animal Care and Use Committee. Mice were housed and mated in a specific pathogen-free facility under the care of the Central Research Department of Rhode Island Hospital. The day of vaginal plug appearance was designated gestational day (gd) 0. On indicated gd, the animals were euthanized. Blood was collected via cardiac puncture for serum preparation. The liver and placenta were snap-frozen in liquid nitrogen and then stored at −80 °C for immunoblotting analysis or fixed in 10% formaldehyde for immunofluorescence staining. For IL-10 administration, recombinant mouse IL-10 (5 ng/μL, dissolved in DPBS, 500 ng/per mouse, n = 5) or an equivalent volume (100 μL) of DPBS (n = 5) as control was injected intraperitoneally into IL-10^−/−^ mice. On day 1 and 2 after injection, the animals were euthanized, and all samples were collected as described above.

The use of sera from age-matched non-pregnant women and women with normal pregnancies was approved by the Institutional Review Boards at Women and Infants Hospital, Providence, RI.

### 2.2. Real-Time Quantitative Polymerase Chain Reaction

Total RNA was extracted from mouse liver and placenta tissues using TRI-reagent (Sigma–Aldrich, St. Louis, MO, USA) followed by DNase treatment with a Turbo DNase kit (Ambion, Austin, TX, USA) to eliminate any genomic DNA contamination. Reverse transcription (RT) was performed using Superscript III reverse transcriptase (Life Technologies, Grand Island, NY, USA) at 50 °C for 1 h. PCR was performed for 35 cycles using the GoTaq Green PCR MasterMix (Promega, Madison, WI, USA) in an Eppendorf Mastercycler Gradient (Eppendorf, Hamburg, Germany). The sequences of the primers for mouse transthyretin were sense, 5′-TGG AAA TCA CAC GGG GA-3′, and antisense, 5′-GCT CCT CGT GAA TCC CTT-3′. The PCR cycling profiles were 94 °C for 30 s, 55 °C for 30 s, and 72 °C for 60 s, with a final extension at 72 °C for 10 min, finishing with a melting curve step. The collected CT values were normalized to internal control, housekeeping gene actin. The obtained values from each group were compared with respect to the values obtained from their respective control groups. There were five determinations using five individual experiments. The results were expressed as the fold change of corresponding genes of the control. The PCR products were visualized using agarose gel electrophoresis following standard procedures.

### 2.3. Serum Preparation and Enzyme-Linked Immunosorbent Assay (ELISA)

Following a resting period of 15 to 30 min at room temperature, blood was centrifuged at 500× *g* for 10 min, and the supernatant was collected and stored at −80 °C until further use. TTR concentration in mouse serum was measured using corresponding ELISA kits according to the manufacturer’s instructions (Mouse TTR ELISA Kit, Innovative Research, Michigan, MI, USA). Serum levels of human transthyretin were measured by using a transthyretin-specific enzyme-linked immunosorbent assay kit (ALPCO kit, Salem, NH, USA), according to the manufacturer’s protocol. Briefly, the serum samples were diluted 1:10,000 or 1:20,000 in wash buffer and incubated on a TTR antibody-precoated plate for 1 h at room temperature, washed, and incubated with peroxidase-labeled TTR antibody for 1 h. Chromogenic reaction was developed by incubating with TMB substrate for 10 min, and the reaction was stopped via the addition of HCl solution. The optical density value was measured at 450 nm with a microplate reader (Bio-Rad, Hercules, CA, USA). Mouse albumin in serum was also measured using ELISA (Mouse DuoSet, R&D Systems Inc., Minneapolis, MN, USA) according to the manufacturer’s instructions.

### 2.4. Immunofluorescence Staining

Formalin-fixed liver and placenta tissues were sectioned at 10 μm and deparaffinized by heating slides to 60 °C for 1 h, followed by three consecutive extractions in Citrisolv (Fisher Scientific, Pittsburgh, PA, USA). Sections were blocked in blocking buffer containing 3% bovine serum albumin and 0.1% Triton X-100 in PBS and then incubated with primary antibodies, rabbit anti-TTR antibody (Agilent Dako, Santa Clara, CA, USA) or mouse anti-cytokeratin 7 (Novus Biologicals, Littleton, CO, USA) overnight at 4 °C. Primary antibody-bound target proteins were visualized with goat anti-mouse Alexa-Fluor 488 or anti-rabbit Alexa-Fluor 594 for 1 h at room temperature. The specificity of anti-transthyretin antibody was confirmed as evidenced by the fact that positive transthyretin staining was blocked either by saturating the primary antibody with mouse serum or by using normal serum IgG instead of the primary antibody. Images were captured with a Nikon Eclipse 80i microscope (Nikon, Inc., Melville, NY, USA) equipped with a Nikon Plan Fluor 100 0.5–1.3 Oil Iris with differential interference contrast and epifluorescent capabilities using a Qimaging Retiga 2000R digital camera and Nikon imaging software (NIS-Elements-BR 3.0, Minato City, Tokyo, Japan). Figures were processed with brightness/contrast adjustment using Photoshop CS2 (Adobe, San Jose, CA, USA).

### 2.5. Immunoblotting

Mouse liver and placenta tissues were homogenized on ice in immunoprecipitation assay-buffered detergent (RIPA) [50 mm Tris (pH 7.6), 1% Triton X-100, 1% deoxycholate, 0.1% SDS, 150 mm NaCl, 50 mm NaF, 2 mm phenylmethylsulfonylfluoride plus protease inhibitors (Complete TM)]. Total cell protein was determined using a BCA protein assay. Samples were fractionated by 4–15% SDS–PAGE and electrotransferred to PVDF membrane. The membrane was blocked with nonfat dried milk (5%) prepared in phosphate-buffered saline containing 0.1% Tween 20 (PBST) for 1 h at RT. Proteins were detected via incubating with rabbit anti-transthyretin antibodies (DAKO) in PBST overnight 4 °C, secondary goat anti-rabbit horseradish peroxidase (GE Healthcare, Chicago, IL, USA) for 1 h at room temperature, and finally visualized by incubating with enhanced chemiluminescence substrate (GE Healthcare, Chicago, IL, USA). The density of the blots was measured with ImageJ (NIH).

### 2.6. Statistical Analysis

Statistical comparisons were conducted using Student’s *t*-test or one-way ANOVA using GraphPad Prism (version 9.3). For one-way ANOVA, a Tukey’s Honestly Significant Difference post hoc test was performed. The results were expressed as the mean ± SEM from at least three independent experiments. *p* values of less than 0.05 were considered statistically significant.

## 3. Results

### 3.1. Temporal Changes in Circulating TTR during Mouse Pregnancy

Using specific ELISA kits, mouse serum samples were assessed for TTR concentration. Its concentration in sera from non-pregnant mice was 548.6 ± 80.5 μg/mL on average. Surprisingly, serum TTR in pregnant mice significantly decreased in a temporal manner through the gestational period. TTR dropped to 432.7 ± 52.0 μg/mL very early during the implantation period on gd 3–5 (*p* < 0.05) and continued to drop through gd 12–14 reaching the concentration of 100 ± 37.5 μg/mL (*p* < 0.05) (Figure 1A). TTR began to increase through gd 17–18 (*p* < 0.05) and continued to rise after delivery with 433.2 ± 57.1 μg/mL on postpartum day 1 (PD1) (Figure 1B).

To determine whether the decline in TTR concentration is specific, we measured the concentration of albumin, a major serum protein. Our ELISA results showed that the levels of albumin were mildly but significantly elevated throughout gestation in comparison to non-pregnant mice (10.8 ± 1.3 μg/mL): 12.2 ± 1.6 μg/mL on gd 3–5 (*p* < 0.05), 14.2 ± 1.6 μg/mL on gd 9–10 (*p* < 0.05), 15.8 ± 3.1 μg/mL on gd 12–14 (*p* < 0.05), 14.8 ± 1.5 μg/mL on gd 17–18 (*p* < 0.05) and 18.9 ± 4.8 μg/mL on postpartum day (PD) 1 (*p* < 0.05) (Figure 1B). Furthermore, the concentration of total serum proteins from non-pregnant and pregnant mice on different gestation days was measured. No significant changes in total serum protein concentrations were detected (Figure 1C, *p* > 0.05).

To determine whether the gestational alteration patterns of TTR, as seen in mice, also occurs in human pregnancy, we measured TTR levels in sera from non-pregnant and pregnant women at weeks 12–14 and term using ELISA kits. The results indicated that TTRs levels on weeks 12–14 significantly decreased compared to non-pregnant women. As seen in pregnant mice, TTR levels at term were restored almost to levels seen in non-pregnant women (Figure 2).

### 3.2. Expression Pattern of TTR in the Mouse Liver at Protein and mRNA Levels during Gestation

Since the liver is the primary source of serum TTR, we next examined TTR expression in mouse liver using Western blotting and immunofluorescence staining for protein levels as well as RT-PCR for mRNA levels. Western blotting showed significantly lower levels of TTR protein content in pregnant mice on gd12, gd14 and gd17 when compared with non-pregnant mice (Figure 3A, *p* < 0.05). The liver TTR protein expression returned to normal levels on day 1 post-partum (Figure 3A,B). Similar results were also observed using immunofluorescent staining (Figure 3C). To determine whether changes in TTR protein expression result from the alterations in TTR mRNA, we isolated total mRNA from the liver on several gestational days and performed RT-PCR. As shown in Figure 3C, alterations in liver TTR mRNA are consistent with those in TTR protein expression on corresponding gestational days.

### 3.3. TTR Protein and mRNA Expression in the Placenta

Implantation units and placental tissues at various gestational days were collected and assessed for TTR expression. Immunoblotting revealed that the mouse placenta presented a pattern of TTR expression that was similar to that observed in the liver (Figure 4A,B). Consistent results were also obtained using immunofluorescence staining (Figure 4C). Additionally, TTR immunoreactivity was mainly concentrated in the junctional zone of the placenta (Figure 4C). RT-PCR analysis showed that placental TTR mRNA levels altered in a way that corresponded to TTR protein presence at gd12, 14, and 17. TTR mRNA was not detectable at gd7 (Figure 4D).

### 3.4. Presence of TTR in Sera from Non-Pregnant and Pregnant IL-10^−/−^ Mice 

IL-10 is a pleiotropic anti-inflammatory cytokine that plays an important role in normal pregnancy [26]. IL-10^−/−^ mice exhibit intrinsic inflammatory phenotype and serve as an animal model of chronic inflammation [27]. Our prior studies have revealed that IL-10 deficiency, coupled with adverse factors such as hypoxia and lipopolysaccharide, is associated with the onset of adverse pregnancy outcomes in mice [28,29]. Since IL-10^−/−^ mice show significantly higher severity index of adverse pregnancy outcomes, we hypothesized that this strain of mice might exhibit downregulated TTR as compared to wild-type counterparts. To address this, we utilized pregnant IL-10^−/−^ mice. First, we compared serum TTR levels between non-pregnant IL-10^−/−^ mice and wild-type mice and then measured the TTR levels in IL-10^−/−^ mice throughout gestation to see if TTR undergoes temporal changes similar to that observed in wild-type mice during pregnancy.

The ELISA results showed significantly reduced levels of serum TTR (357.8 ± 23.6 µg/mL) in non-pregnant IL-10^−/−^ mice compared to non-pregnant wild-type mice (Figure 5A). This result suggests a regulatory role for IL-10 in TTR expression. We then tested whether administration of recombinant IL-10 would restore TTR levels in IL-10^−/−^ mice. We administrated IL-10 or vehicle to non-pregnant IL-10^−/−^ mice and measured TTR levels on day 1 and 2 after injection. TTR levels increased to 579.6 ± 19.2 µg/mL on day 1 and further increased to 690 ± 114.8 µg/mL on day 2 after IL-10 injection compared to the controls (395.2 ± 19.2 µg/mL) (Figure 5B). However, vehicle-only treatment failed to increase blood TTR (Figure 5B).

Next, we detected serum TTR levels in IL-10^−/−^ mice during pregnancy and found that the animals exhibited a progressive decline in serum TTR levels throughout gestation, with the lowest levels at gd 16–17 (Figure 5C). However, at day 1 post-partum, TTR rose to levels comparable to those observed prior to pregnancy.

### 3.5. Overexpression of Human TTR in Transgenic Mice Disturbed Normal Pregnancy

TTR is prone to forming toxic aggregates, especially in an inflammatory environment, and cells have limited capacity to process protein aggregates through the autophagy–lysosomal machinery. Thus, apparently, a high level of TTR will result in more aggregates, which may not be easily degraded in a time-effective manner. Intriguingly, our above results demonstrated a remarkably low level of TTR presence during pregnancy, especially at gd 12–14. A low level of TTR, such as 100 ug/mL, may form fewer protein aggregates during pregnancy, which can be easily cleared. Thus, we speculate that this downregulation of TTR presence during pregnancy is beneficial for maintaining a normal pregnancy. If this is true, increasing TTR presence during pregnancy may interfere with normal pregnancy. To address this, we utilized a strain of transgenic (huTTR) mice that contains 90–100 copies of the wild-type human TTR gene with all its known regulatory elements and examined the effect of higher-than-normal levels of TTR on pregnancy outcomes, including implantation units, fetal weight, and litter size. First, we measured the serum concentration of huTTR. Since these mice develop amorphous and fibrillar TTR tissue deposits after one year of age, we only studied pregnant mice that were 7–9 weeks of age. ELISA measurement showed high levels of human TTR (113 mg/dl on average) in sera from non-pregnant huTTR mice. As shown in Figure 6, huTTR mice exhibited a lower number of embryo units, reduced fetal weight, and smaller litter size compared to wild-type mice.

## 4. Discussion

In this study, we investigated the temporal expression of TTR in serum and the placenta during pregnancy. Our findings indicate a substantial reduction in TTR levels during the middle of pregnancy, followed by restoration to normal levels at term and post-partum. Furthermore, pregnancy-compatible cytokine IL-10 plays a key role in maintaining TTR levels at the wild-type levels as IL-10^−/−^ mice have much lower levels of TTR and IL-10 administration restores these levels. Notably, a similar temporal decline pattern of TTR was observed in pregnant women. It appears that temporal regulation of TTR is a normal occurrence in pregnancy. This is also reflected in the overall TTR levels in the mouse liver during pregnancy. Curiously, transgenic overexpression of human TTR interferes with normal pregnancy, as evidenced by decreased embryo units, fetal weight, and litter size.

Serum TTR is primarily released from the liver, the main organ responsible for TTR synthesis. Inflammation or infection can dramatically inhibit the production and release of TTR in the liver, leading to a decrease in serum TTR levels [30,31]. Consequently, TTR is regarded as a negative acute phase protein in the liver [30,31]. Interestingly, during the implantation process of a fertilized egg, a physiological inflammatory microenvironment in the uterus is required [23,24,32]. To determine whether sterile inflammation may be responsible for the downregulation of TTR production during pregnancy, we utilized IL-10^−/−^ mice that exhibit constitutive inflammation due to the lack of IL-10, a strong anti-inflammatory cytokine. We compared the serum levels of TTR between non-pregnant IL-10^−/−^ and wild-type mice and investigated the effect of IL-10 deficiency on serum levels of TTR across gestation. Our results showed that IL-10^−/−^ mice exhibited a significantly lower level of TTR than wild-type mice, and the administration of recombinant IL-10 restored the TTR concentration in IL-10^−/−^ mice. Furthermore, like wild-type mice, IL-10^−/−^ mice also displayed a progressive drop in TTR concentration over pregnancy. However, the peak of TTR decline in IL-10^−/−^ mice occurred at gd17, which contrasted with gd12–14 in wild-type mice. These findings support the notion that inflammation may negatively impact TTR presence in sera during pregnancy. Therefore, we postulated that pregnancy-associated inflammation may trigger the inhibition of TTR production and release from the liver. However, whether IL-10 itself can increase TTR production by directly acting on cells in the liver or placenta and whether the loss of IL-10 may, in part, contribute to decreased TTR abundance remains to be addressed. Prior studies have revealed that PE patients have higher levels of inflammation. Hence, the enhanced inflammation in PE women may contribute to lower levels of serum TTR presence in PE patients at term that we reported before [16,23,24,25]. This could also be associated with the aggregated phenotype of TTR as detected in PE, which may further alter TTR regulation [16].

Downregulation of TTR during pregnancy may benefit the maintenance of normal pregnancy since TTR is prone to form cytotoxic aggregates under adverse conditions such as chronic hypoxia and inflammation. Thus, we hypothesized that the downregulation of TTR may be an adaptive response, preventing TTR misfolding and aggregation under an inflammatory microenvironment at the maternal–fetal interface. To validate this, we used transgenic mice that overexpress human TTR with serum concentration at around 113 mg/dl and investigated its effect on reproductive outcomes. Our results indicated that transgenic mice exhibited reduced implantation units, lower fetal weight, and smaller litter size compared to wild-type mice. This suggests that higher-than-normal levels of TTR presence in the sera during pregnancy interfere with normal reproductive processes. In fact, our prior studies have demonstrated that human TTR-overexpressing transgenic mice manifest a robust deposition of TTR aggregates in the placenta and PE-like features, including hypertension, proteinuria, and the release of anti-angiogenic factors [15]. In contrast, our prior study demonstrates that TTR^−/−^ mice do not differ from wild-type mice with respect to fertility, litter size, and fetal weight.

Given that TTR is one of the major plasma proteins, and its decline in concentration in the serum during pregnancy may alter plasma osmolarity, we measured the concentrations of serum albumin at various gestational days. Our ELISA analysis revealed that serum albumin slowly increased in a gestational day-dependent manner. It has been reported that albumin can also carry thyroxine. Therefore, the elevated albumin may be the compensatory response to changes in blood osmolarity and transport of thyroxine that is negatively affected by serum TTR downregulation.

## 5. Conclusions

Taken together, our findings suggest that the regulation of TTR protein during pregnancy content is temporally programmed, and its regulated downregulation is essential for the maintenance of normal pregnancy, and pregnancy-mediated inflammation may contribute to its downregulation.

## Figures and Tables

**Figure 1 biology-12-01048-f001:**
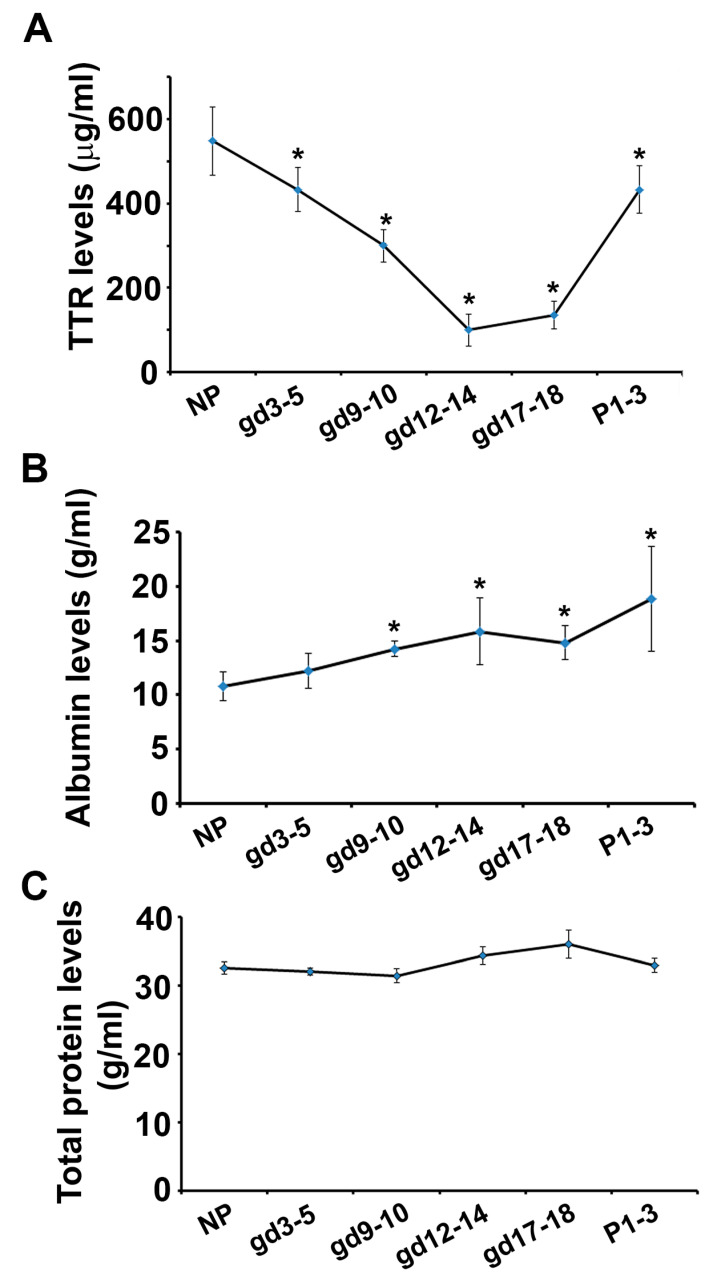
Alterations in the concentrations of TTR, albumin, and total proteins in sera from pregnant and age-matched non-pregnant mice. (**A**,**B**), TTR (**A**) and albumin (**B**) concentrations were measured in mouse sera at indicated time points using respective ELISA kits. (**C**) The concentration of total serum proteins was measured using BCA method. Data were expressed as mean ± SEM and statistically analyzed via one-way ANOVA (n = 8) *: *p* < 0.05.

**Figure 2 biology-12-01048-f002:**
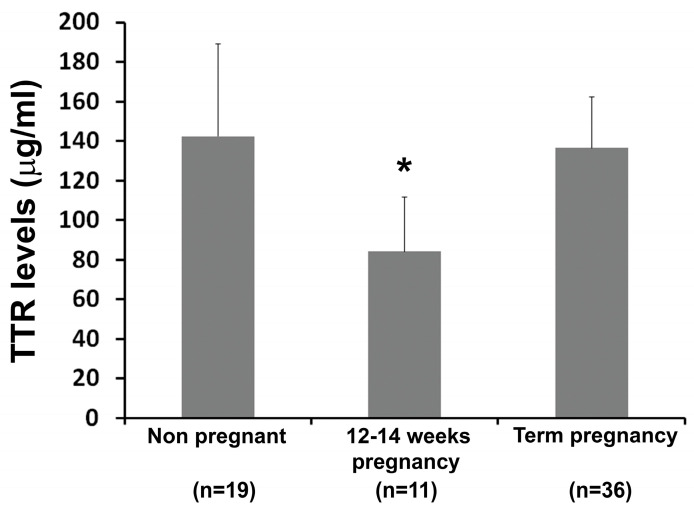
TTR concentration in sera from pregnant women at term and 12–14 weeks of gestation and age-matched non-pregnant women. TTR concentration in sera at 12–14 weeks of gestation was lower than that in non-pregnancy and term pregnancy. Data were expressed as mean ± SEM and statistically analyzed via one-way ANOVA. *: *p* < 0.05.

**Figure 3 biology-12-01048-f003:**
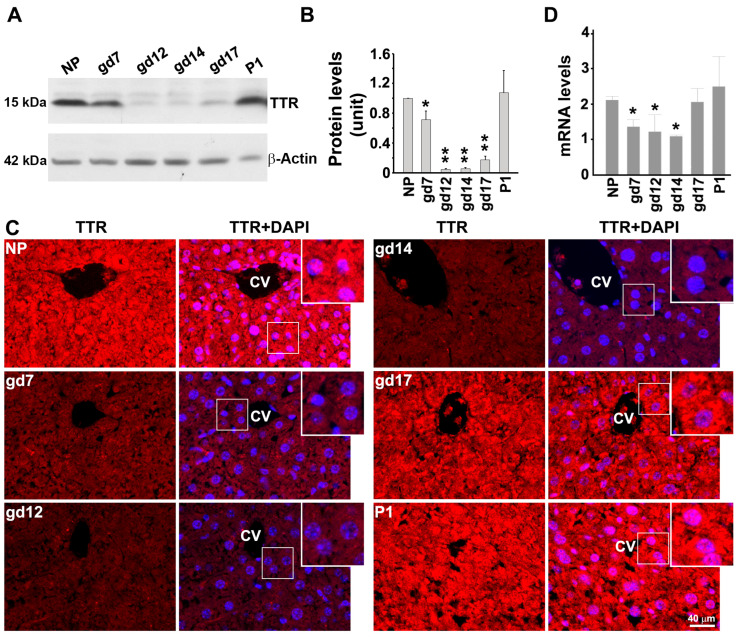
The levels of TTR protein and mRNA in the livers of non-pregnant (NP) and pregnant mice at various gestational days (gd) and day 1 postpartum (P1). (**A**), Live tissues at different time points were subject to Western blotting analysis (See Appendix A for the original full WB figures). (**B**), The intensity of TTR bands was quantified and statistically analyzed. (**C**), Liver sections were stained with anti-TTR antibody (red). The nuclei were stained with DAPI (blue). Inserts were magnified images of boxed areas. CV: central vein. Scale bar: 40 μm. (**D**), Comparison of TTR mRNA levels of the livers of non-pregnant (NP) and pregnant mice at various time points. Data were expressed as mean ± SEM and statistically analyzed via one-way ANOVA. n = 4; *: *p* < 0.05; **: *p* < 0.01.

**Figure 4 biology-12-01048-f004:**
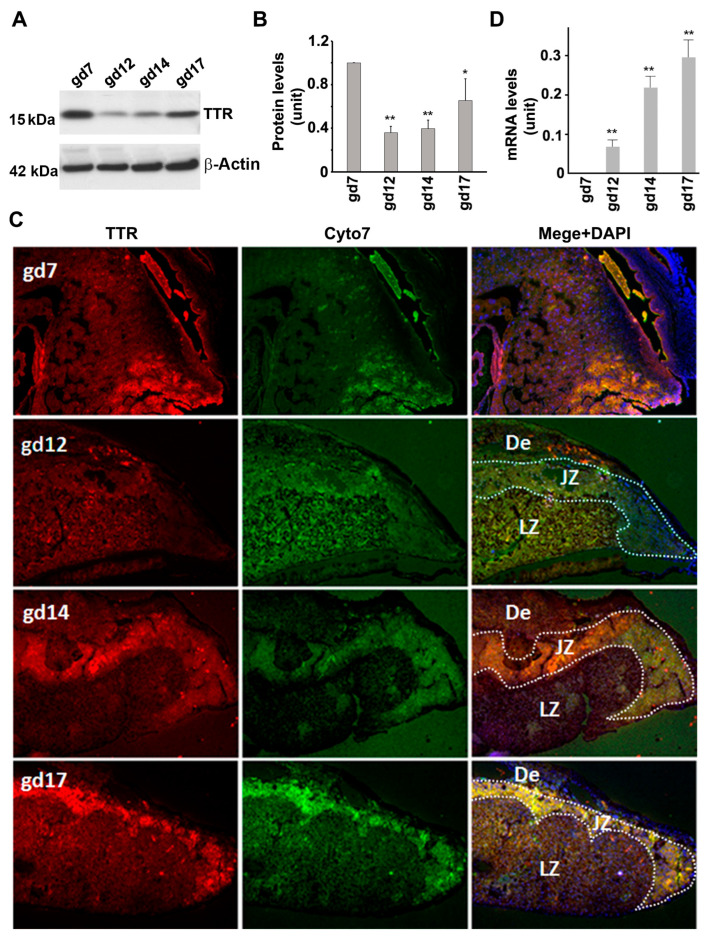
TTR expression at protein and mRNA levels in the placenta of mice during pregnancy. (**A**,**B**), Western blotting and quantitative data show TTR protein expression in the placenta of mice at various gestational days (See Appendix A for the original full WB figures). Data were expressed as mean ± SEM and statistically analyzed via one-way ANOVA. n = 4; *: *p* < 0.05; **: *p* < 0.01. (**C**), Immunofluorescence staining demonstrates the distribution of TTR (red) and cytokeratin 7 (Cyto7, green) in the placenta at different gestational days. The nuclei were stained with DAPI (blue). (**D**), TTR mRNA expression in the placenta of mice at various time points. De: decidua; JZ: junctional zone: LZ: labyrinth zone. Scale bar: 40 μm.

**Figure 5 biology-12-01048-f005:**
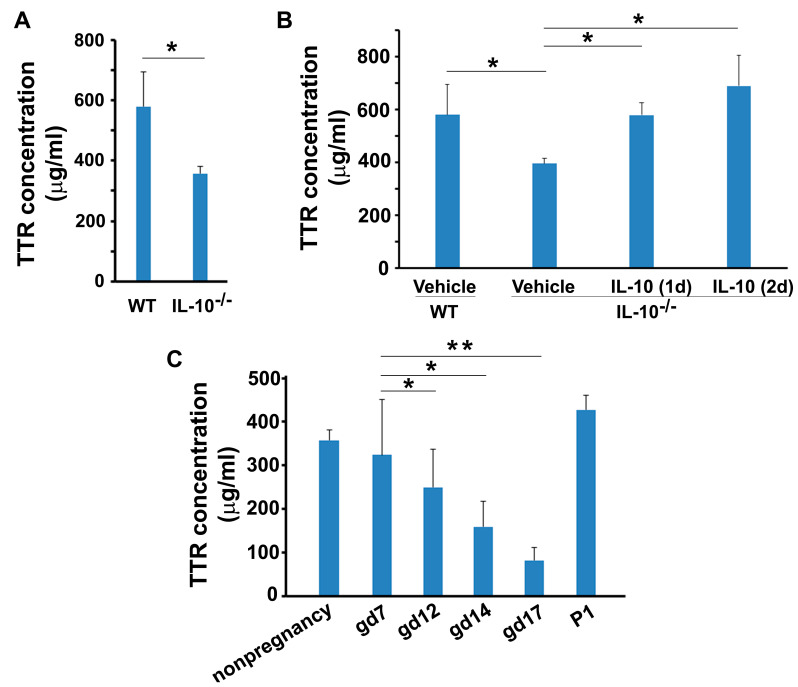
The effect of IL-10 on serum TTR concentration in non-pregnant and pregnant mice. (**A**), Comparison of TTR serum concentration between IL-10^−/−^ and wild-type mice. (**B**), Administration of IL-10 in IL-10^−/−^ mice restore the level of TTR in sera. TTR concentration was measured in sera from wild-type mice with vehicle injection and IL-10^−/−^ mice with vehicle or IL-10 injection for 1 day or 2 days. (**C**), Alterations in serum TTR concentration in non-pregnant IL-10^−/−^ mice and pregnant IL-10^−/−^ mice at various gestational days. Data were expressed as mean ± SEM and statistically analyzed via one-way ANOVA. *: *p* < 0.05; **: *p* < 0.01, n = 5.

**Figure 6 biology-12-01048-f006:**
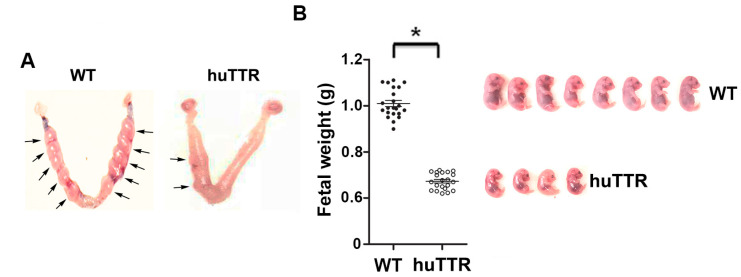
Transgenic mice that overexpress TTR exhibit defective reproductive outcomes. (**A**), A representative set for comparison of embryo units between wild-type (WT) and transgenic mice (TrTTR) at gd 9. Arrows indicate embryo units. (**B**), Comparison of fetal weight between wild-type (WT) and transgenic mice (huTTR) gd 17. Fetal images show a representative litter of WT and huTTR pregnant mice. Data were expressed as mean ± SEM and statistically analyzed by a student-*t* test. n = 22; *: *p* < 0.01.

## Data Availability

Data are contained within the article.

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
