# Peer review of "Gestational Age-Dependent Regulation of Transthyretin in Mice during Pregnancy"

_biology, 2023, doi:10.3390/biology12081048_

Round 1

Reviewer 1 Report

The manuscript entitled “Gestational age-dependent regulation of transthyretin in mice during pregnancy” evaluated the serum, liver and placental profile of TTR in mice throughout pregnancy, as well as in IL-10−/−mice and that over-expressing human TTR (huTTR mice). The findings are very interesting, the methodology is adequate for the proposed objectives and the discussion of the results is well done. However, some corrections in the manuscript are needed.

Line 85: Inform the number of animals used.

Lines 110-111: Inform the housekeeping gene used and by which method the gene analysis was performed.

Lines 133-134; Lines 152: Inform the specifications of the antibodies used.

Lines 157; 185; 197; 220; 234; 271: Inform the statistical test used in the ANOVA.

Line 186: It is not appropriate to say “dysregulated”, since it is a physiological reduction.

Line 190: “TTR levels were restored almost to the levels seen in non-pregnant women”. In which gestational period? Inform that it was during the term pregnancy.

Line 232: “quantitative day”?

Line 238: Remove “during gestation”

Line 294: “PES”?

Author Response

July 20, 2023

Dear editors

We thank the Editors and the Reviewers for their thoughtful comments on our manuscript, entitled “Gestational age-dependent regulation of transthyretin in mice during pregnancy”. Below, we provide a point-by-point response to the Reviewers’ comments. As suggested, we have provided a copy with tracked changes.  We hope our manuscript is now suitable for publication in Biology.

All authors concur with the submission of the revised manuscript.

Sincerely,

Surendra Sharma, MD, PhD

Professor of Pediatrics

Women and Infants Hospital-Brown University

Providence, RI, USA

Shibin Cheng, MD, MS, PhD

Associate Professor of Pediatrics

Women and Infants Hospital-Brown University

Providence, RI, USA

Response to the Reviewers’ comments

Reviewer #1

The manuscript entitled “Gestational age-dependent regulation of transthyretin in mice during pregnancy” evaluated the serum, liver and placental profile of TTR in mice throughout pregnancy, as well as in IL-10−/−mice and that over-expressing human TTR (huTTR mice). The findings are very interesting, the methodology is adequate for the proposed objectives and the discussion of the results is well done. However, some corrections in the manuscript are needed.

Response: We thank the reviewer for the comments.

Line 85: Inform the number of animals used.

Response: We have added the number (See lines 85, 87).

Lines 110-111: Inform the housekeeping gene used and by which method the gene analysis was performed.

Response: We have added suggested contents (See lines 114-118).

Lines 133-134; Lines 152: Inform the specifications of the antibodies used.

Response: We added primary antibodies and secondary antibodies (See lines 140-142).

Lines 157; 185; 197; 220; 234; 271: Inform the statistical test used in the ANOVA.

Response: We have added it (See lines 165, 166).

Line 186: It is not appropriate to say “dysregulated”, since it is a physiological reduction.

Response: “dysregulated” has been replaced by “alteration” (See line 194).

Line 190: “TTR levels were restored almost to the levels seen in non-pregnant women”. In which gestational period? Inform that it was during the term pregnancy.

Response: “at term” has been added as suggested (See line 198).

Line 232: “quantitative day”?

Response: “day” has been corrected to “data” (See line 240).

Line 238: Remove “during gestation”

Response: It has been removed (See line 246).

Line 294: “PES”?

Response: The sentences containing “PES” were deleted as these were confusing, which was also raised by another reviewer (See lines 313-314).

Reviewer 2 Report

See attachment file

Author Response

July 20, 2023

Dear editors

We thank the Editors and the Reviewers for their thoughtful comments on our manuscript, entitled “Gestational age-dependent regulation of transthyretin in mice during pregnancy”. Below, we provide a point-by-point response to the Reviewers’ comments. As suggested, we have provided a copy with tracked changes.  We hope our manuscript is now suitable for publication in Biology.

All authors concur with the submission of the revised manuscript.

Sincerely,

Surendra Sharma, MD, PhD

Professor of Pediatrics

Women and Infants Hospital-Brown University

Providence, RI, USA

Shibin Cheng, MD, MS, PhD

Associate Professor of Pediatrics

Women and Infants Hospital-Brown University

Providence, RI, USA

Response to the Reviewers’ comments

Reviewer #2

This manuscript can be evaluated as a useful article with effective experiments to evaluate the role of TTR in abnormal pregnancy; the variation of blood levels of TTR was a very interesting result. On the other hand, I would like to address several major comments and minor ones on the content of the manuscript to make it more complete and useful.

Response: We thank the reviewer for the comments.

Major comments

  • Additional explanation of the purpose and design of the experiments using human TTR-Tg mice seemed to be

Response: We have reedited the description regarding the rationale of the experiments using TTR-Tg mice (See lines 282-291).

  1. L282-283 "high levels of human TTR (113 mg/dl on average) in sera" and L338 “higher-than-normal levels of TTR”

Are they really at a high level? In Fig. 1A, mouse TTR showed 400 ug/mL. The authors should show the data of human TTR in the control group of mice (non-Tg mice) to confirm the representation.

Response: As for human TTR concentration in sera of TTR-Tg mice, 113 mg/dl = 1130 ug/ml, which is way higher than mouse TTR concentration in non-pregnant mice and pregnant mice (400 ug/ml) at gd7.

Non-TTR-Tg mice do not express human TTR.

  1. L336-339 " Our results indicated that  "

Did the 113 mg/mL of human TTR imply an AGGREGATION type? What was the difference of effects to the 100 ug/mL mouse TTR (gd12-14) that did not cause pregnancy interference in the normal mouse pregnancy?

Response: TTR is prone to forming toxic aggregates, especially in inflammatory environment, and cells have limited capacity to process protein aggregates through the autophagy-lysosomal machinery. Thus, apparently, a high level of TTR will result in more aggregates, which may not be easily degraded in a time-effective manner. This may be the reason why we observed a large amount of human TTR aggregates in the placenta from TTR-Tg mice with an average of 113 mg/dl (1130 ug/ml) of human TTR in circulation (See Reference 17). In wild-type mice, mouse TTR dropped to 100 ug/ml at gd 12-14, which is way lower than that (500 ug/ml) in non-pregnant mice. A low level of TTR such as 100 ug/ml may form fewer protein aggregates during pregnancy, which can be easily cleared. Thus, we speculate that this downregulation of TTR presence during pregnancy is beneficial for maintaining a normal pregnancy. If this is true, increasing TTR presence during pregnancy may interfere with the normal pregnancy. To test this, we used transgenic mice overexpressing human TTR and examined the effect of higher-than-normal levels of TTR on pregnancy outcomes. As expected, transgenic mice exhibited adverse pregnancy complications (See lines 282-291).

  • L294 "Our previous work has suggested that TTR aggregation induced by "

The authors wrote that IL-10 deficiency is involved in hypoxia and pregnancy failure (L241-243), but not that hypoxia induces TTR aggregation. If the authors want this sentence to use as the basis for discussion, it should be referred to beforehand in the Introduction, and the hypothesis of L294 should be explained logically.

Response: We did not mean IL-10 deficiency is involved in hypoxia and pregnancy failure. We meant that IL-10 deficiency combined with hypoxia treatment can lead to adverse pregnancy complications. To avoid possible confusion, we removed this description (See lines 313-315).

The authors wrote the sentence in L308-309 that inflammation suppresses TTR expression in the liver. However, in L323-324, they wrote that TTR induces the situation of “pregnancy-associated inflammation”. These two statements seem to contradict each other, so please resolve them.   

Response: We did not mean that TTR induces the situation of “pregnancy-associated inflammation”. Instead, normal pregnancy condition induces mild sterile inflammation as compared to non-pregnancy condition. We hypothesized that normal pregnancy-induced inflammation may contribute to downregulation of TTR abundance during pregnancy.

Also, is there any reason that the authors did not propose a mechanism that IL-10 acts directly on cells in the liver or placenta to increase TTR? (This hypothesis includes IL-10 can induce without involving inflammation)

Response: This is a good question. We could not exclude the possibility that IL-10 directly acts on cells in the liver or placenta and induces TTR production and release. Thus, the loss of IL-10 itself may, in part, contribute to decreased TTR abundance in addition to chronic inflammation in IL-10 knockout mice. We have added this description in the Discussion section (See lines 344-346).

  • L135: "anti-mouse Alexa-Fluor 594"

  Did you perform any kind of “Mouse on Mouse treatment”? If this anti-mouse antibody is used on mouse tissue without specific treatments, a strong non-specific reaction always happens. Or are these descriptions of the secondary antibodies correct?

Response: We thank the reviewer for pointing out this error. We have added the primary antibodies and their corresponding secondary antibodies (See lines 140-142).

Minor comments

L60: "if PES was depleted of aggregated TTR"   

   (an example of correction) "if aggregated TTR was depleted from PES-treated mice"

 Response: We have modified the description as suggested.

L110-111: "50 -TGGAAA TCA CAC GGG GA-30, and antisense, 50 -GCT CCT CGT GAA TCC CTT-30"   

   "5'-TGGAAA TCA CAC GGG GA-3', and antisense, 5'-GCT CCT CGT GAA TCC CTT- 3 '"

 Response: We have corrected this typo (See lines 111, 112).

2.4. Immunofluorescence staining: Please specify the primary antibody you used.

 Response: We have added this information (See lines 140, 141).

L203: "Figure 3a"     "Figure 3A"

L206: "Figure 3B"    "Figure 3C"

L247 "Sine"    "Since"

Response: We have corrected these (See lines 211, 212, 213, and 251).

L257-258 "5B). However, no obvious changes in TTR levels were observed in vehicle-injected animals (data not shown)."

“data not shown" is incorrect. Results are shown in Figure 5B. “no obvious changes” is also incorrect. There are significant differences in the TTR levels between vehicle-injected WT mice and vehicle-injected IL-10-/- mice.

   (An example of a modification) "However, vehicle-only treatment failed to increase blood TTR (Figure 5B)."

Response: We have re-edited the description accordingly (See lines 265-267).
